# Reduced Prognostic Role of Serum PCT Measurement in Very Frail Older Adults Admitted to the Emergency Department

**DOI:** 10.3390/antibiotics12061036

**Published:** 2023-06-10

**Authors:** Andrea Russo, Sara Salini, Giordana Gava, Giuseppe Merra, Andrea Piccioni, Giuseppe De Matteis, Gianluca Tullo, Angela Novelli, Martina Petrucci, Antonio Gasbarrini, Francesco Landi, Francesco Franceschi, Marcello Covino

**Affiliations:** 1Department of Geriatrics, Fondazione Policlinico Universitario A. Gemelli, IRCCS, 00168 Rome, Italy; andrea.russo1@policlinicogemelli.it (A.R.); francesco.landi@unicatt.it (F.L.); 2Medicine and Surgery, Università Cattolica del Sacro Cuore, 00168 Rome, Italy; giordana.gava@gmail.com (G.G.); antonio.gasbarrini@policlinicogemelli.it (A.G.); francesco.franceschi@policlinicogemelli.it (F.F.); marcello.covino@policlinicogemelli.it (M.C.); 3Department of Biomedicine and Prevention, University of Rome Tor Vergata, 00133 Rome, Italy; giuseppe.merra@uniroma2.it; 4Emergency Department, Fondazione Policlinico Universitario A. Gemelli, IRCCS, 00168 Rome, Italy; andrea.piccioni@policlinicogemelli.it (A.P.); gianlucatullo@gmail.com (G.T.); angela.novelli@policlinicogemelli.it (A.N.); martina.petrucci@policlinicogemelli.it (M.P.); 5Department of Internal Medicina and Gastroenterology, Fondazione Policlinico Universitario A. Gemelli, IRCCS, 00168 Rome, Italy; giuseppe.dematteis@policlinicogemelli.it

**Keywords:** procalcitonin, elderly, frailty, emergency department, sepsis

## Abstract

Background: This study aims to evaluate the prognostic role of serum PCT in older patients with suspect sepsis or infective diagnosis in the Emergency Department (ED) with a particular focus on the clinical consequences and characteristics due to frailty status. Methods: This is a observational retrospective study conducted in the ED of a teaching hospital. We identified all consecutive patients aged ≥ 80 years admitted to the ED and subsequently hospitalized for clinical suspicion of infection. Inclusion criteria were: age ≥ 80 years and clinical suspicion of infection; availability of a PCT determination obtained < 24 h since ED access; and Clinical Frailty Scale (CFS) determination. Study endpoints were the diagnostic accuracy of PCT for all-cause in-hospital death, infective diagnosis at discharge, and bloodstream infection. Diagnostic accuracy was calculated via ROC analysis and compared in the patients with severe frailty, measured by CFS > 6, and patients with low or moderate frailty (CFS 1–6). A multivariate analysis was performed to calculate the adjusted odds of raised PCT values for the study endpoints. Results: In total, 1459 adults ≥ 80 years with a clinical suspicion of infection were included in the study cohort. The median age of the sample was 85 years (82–89), with 718 (49.2%) males. The multivariate models revealed that, after adjusting for significant covariates, the PCT values at ED admission were significantly associated with higher odds of infective diagnosis only in the fit/moderately frail group (Odds Ratio [95% CI] 1.04 [1.01–1.08], *p* 0.009) and not in very frail patients (Odds Ratio [95% CI] 1.02 [0.99–1.06], *p* 0.130). Similarly, PCT values were significantly associated with higher odds of in-hospital death in the fit/moderately frail group (Odds Ratio [95% CI] 1.01 [1.00–1.02], *p* 0.047), but not in the very frail ones (Odds Ratio [95% CI] 1.00 [0.98–1.02], *p* 0.948). Conversely, the PCT values were confirmed to be a good independent predictor of bloodstream infection in both the fit/moderately frail group (Odds Ratio [95% CI] 1.06 [1.04–1.08], *p* < 0.001) and the very frail group (Odds Ratio [95% CI] 1.05 [1.03–1.07], *p* < 0.001). Conclusions: The PCT values at ED admission do not predict infective diagnosis, nor are associated with higher odds of in-hospital death. Still, in frail older adults, the PCT values in ED could be a useful predictor of bloodstream infection.

## 1. Introduction

The world’s population is rapidly aging, with the proportion of older adults (aged 65 years and older) expected to increase from 9% in 2020 to 16% in 2050. The aging of the population has significant implications for healthcare, as older adults have a higher prevalence of chronic diseases, functional impairment, and cognitive decline, which increase their risk of acute illness and emergency department (ED) admission [1,2,3]. Sepsis is a life-threatening organ dysfunction caused by a dysregulated host response to infection [4]. A certain diagnosis of sepsis can be obtained via microbiological tests; however, their results are not readily available in EDs [5,6]. In clinical practice, the quick sequential organ failure assessment score (qSOFA) is often used as a tool for the early diagnosis of septic patients [4]. Furthermore, in these patients, the use of procalcitonin (PCT) as a biomarker can predict the prognosis and guide antibiotic therapy [7], and can be used to stratify the prognostic risk since ED admission [8,9,10].

While the use of qSOFA and PCT can allow the early detection of bloodstream infection and sepsis and guide the introduction of prompt empirical antibiotics and medical supportive therapy [11], this may not be true for older adults, particularly in patients ≥ 80 years, in which a diagnosis of sepsis may be very elusive.

There is growing evidence of a bidirectional correlation between frailty and inflammation in very elderly patients, although the mechanism has not yet been clarified [12]. This correlation similarly impacts markers of infection/inflammation [13,14].

Indeed, these patients often develop serious infections without displaying a typical symptomatic pattern, such as fever, chills, or signs/symptoms of organ involvement, either because of the anergy of the elderly immune system or because of the presence of comorbidities that alter the local and systemic response to infection [15,16,17,18].

The peculiar characteristics of older patients can be resumed by the definition of frailty, which identifies their decreased physiological reserves, increased vulnerability to stressors, and increased risk of adverse outcomes, such as falls, disability, hospitalization, and mortality [19]. Moreover, frailty is associated with a higher prevalence of acute and chronic illnesses, including infections, which are a leading cause of hospitalization and mortality in this population [3].

This study aims to evaluate the prognostic role of the serum PCT and qSOFA score in older patients with suspect sepsis or infective diagnosis in the ED with a particular focus on the clinical consequences and characteristics due to frailty status.

## 2. Results

In the study period, 1459 adults ≥ 80 years with a clinical suspicion of infection were included in the study cohort. The median age of the sample was 85 years (82–89), and 718 (49.2%) patients were male.

Table 1 summarizes the characteristics of patients divided according to frailty patterns. The fit/moderately frail group consisted of 796 (54.6%) patients, and the very frail group of 663 (45.4%) patients. Very frail patients were older and more prevalent in the female sex (*p* < 0.001). The most frequent symptoms recorded in both groups included fever, dyspnea, abdominal pain, and malaise/fatigue. As partly expected, frail patients had more comorbidities and were more frequently admitted for emergency/urgent conditions. In frail patients, the serum PCT at the ED admission was slightly but significantly higher (Table 1). Consistently, the qSOFA at admission was ≥2 in 75 (11.3%) frail patients and in 61 (7.7%) controls (*p* = 0.017).

Overall, 1196 (82.0%) patients had a confirmed diagnosis of infection at hospital discharge. The most represented site of infection in all groups was pneumonia (43.7%), followed by sepsis (22.9%) and urinary tract infections (20.4%) (Table 1).

In-hospital death occurred in 354 (24.3%) cases in the study cohort.

### 2.1. Infection Diagnosis According to Frailty Status

Overall, frail patients had confirmed sepsis in 174/663 (26.2%) cases, compared to 160/796 (20.1%) in fit/moderately frail controls (*p* = 0.005).

An infective diagnosis was confirmed in 568/663 (85.7%) frail patients, compared to 628/796 (78.9%) in fit/moderately frail controls (*p* = 0.001). Stratifying the analysis for frailty, a qSOFA ≥ 2 was associated with infection diagnosis in the fit/moderately frail group, but not in the frail group (Table 2). However, the PCT level was still higher in the case of infections both in the frail group and in the controls (Table 2).

The ROC analysis revealed that the overall discrimination of the PCT value for infections was fair, with ROC AUC 0.626 [0.601–0.651]. PCT diagnostic accuracy was particularly effective for bloodstream infection diagnosis, both in frail patients and in controls (Figure 1).

However, the accuracy for any infective diagnosis was generally lower, mostly in the frail group (ROC AUC 0.634 [0.599–0.667] vs. ROC AUC 0.607 [0.569–0.645], *p* = 0.491 for the comparison) (Figure 2).

### 2.2. All-Cause Death According to Frailty Status

Overall, 216 (32.6%) patients died in the frail group, compared to 138 (17.3%) in the controls (*p* < 0.001). Stratifying the analysis for the frailty group, the factors associated with all-cause death were similar in the two groups (Table 3).

The PCT values were higher in deceased patients compared to those discharged alive in both groups. PCT at admission had a fair discrimination for in-hospital death (ROC AUC 0.620 [0.595–0.645]). Similar to infective diagnosis, the discrimination was higher for the fit/moderately frail group compared to the frail group (ROC AUC 0.659 [0.625–0.692] vs. 0.586 [0.548–0.624], *p* = 0.028) (Figure 3).

### 2.3. Adjusted Odds for Death and Infection Diagnosis According to Frailty

The multivariate models revealed that, after adjusting for significant covariates, the PCT values at the ED admission were significantly associated with higher odds of infective diagnosis only in the fit/moderately frail group (Table 4). Conversely, the PCT values were confirmed to be a good independent predictor of bloodstream infection both in the fit/moderately frail group and very frail group (Table 4).

On the other hand, PCT values were slightly but significantly associated with higher odds of in-hospital death in the fit/moderately frail group, but not in the frail group. In these latter patients, only the severity of clinical conditions at admission were independently associated with higher odds of in-hospital death (Table 4).

## 3. Materials and Methods

### 3.1. Study Design

This is a observational retrospective study conducted in the ED of a teaching hospital with an annual attendance of about 75,000 patients, 87% adults. We identified all consecutive geriatric patients who were admitted to the ED from January 2014 to December 2019 and then hospitalized.

Inclusion criteria were: patients with age ≥ 80 years and presence of clinical signs and symptoms suggestive of bacterial infection; availability of a PCT determination obtained <24 h since ED access; an available multidimensional geriatric assessment; and a qSOFA determination at the first emergency room visit.

Exclusion criteria were age < 80 years.

### 3.2. Study Variables

For all patients included in the study cohort, demographic characteristics (age and sex) and symptoms of admission to the emergency department (fever, dyspnea, chest pain, vomiting, diarrhea, abdominal pain, confusion, and malaise) were reported.

Vital parameters were obtained at ED admission. In the case of several measurements, the first values were considered. For each patient, the qSOFA score on admission was assessed. Frailty was assessed based on the Clinical Frailty Scale (CFS) [20], which is a clinical scale with 9 levels representing different degrees of frailty from 1 (very fit) to 9 (terminally ill).

For all patients, the CFS score was calculated during a multidimensional geriatric assessment performed by a dedicated geriatric unit in the ED. According to the CFS value, patients were divided into two groups: fit/moderately frail for CFS ≤ 6 and frail for CFS > 6.

Comorbidities were recorded, including hypertension, ischemic heart disease, heart failure, chronic respiratory obstructive disease (COPD), peripheral vascular disease, dementia, diabetes, chronic kidney disease, malignancy, and leukemia/lymphoma. Overall comorbidity status was assessed with the Charlson comorbidity index [18].

All patients underwent blood sampling for routine laboratory testing and PCT value determination. Blood and urine cultures were eventually obtained based on the emergency physician’s judgment. The cut-off value of PCT serum level predictive of infection was set at 0.5 ng/mL. All testing was available 24 h a day in our ED. For any patient, where possible, the site of infection was specified: sepsis/bloodstream infection, pneumonia, urinary tract infection (UTI), abdominal infection, or infection in another site.

### 3.3. Outcome Measures

The primary endpoint of the study was the predictive ability of PCT for in-hospital mortality.

As secondary outcomes, we evaluated the occurrence of bloodstream infection and the presence of any infective diagnosis confirmed at the hospital discharge.

The diagnosis of bloodstream infection was defined as the direct isolation of bacteria from two blood samples. Apart from bloodstream infections, the presence of infective diagnosis was supported by clinical examination and the presence of organ-specific symptoms (presented in the tables) associated with direct isolation in any biological specimen, radiological images consistent with infection, and presence of purulent discharge at drainage from any site. We considered the infective diagnoses present or suspected at the time of ED admission. Those diagnosed after more than 7 days since ED admission were considered as developed in-hospital and excluded from the analysis.

### 3.4. Statistical Analysis

Categorical variables are presented as absolute numbers and percentages; continuous variables are presented as median (interquartile range). Categorical variables were statistically compared with Chi-square test or Fisher’s exact test as appropriate. Continuous variables were compared with the Mann–Whitney U test. Significant factors at univariate analysis were entered into a logistic regression model to identify independent risk predictors for the defined outcomes. The PCT value was forced in all the models. When composite variables were entered into the model, the single composing items were excluded to avoid model overfitting and overestimation of the parameters (i.e., the Charlson index excluded the single calculated comorbidities). Receiver operating characteristics (ROC) curve analysis was used to assess the relationship between different PCT values and all-cause in-hospital death and infective diagnosis, and to determine the sensitivity and specificity for the defined outcome at different PCT cut-off levels. ROC curves were compared by the DeLong method for correlated samples and by Z statistic for uncorrelated samples.

All *p* values were 2-sided, with a significance threshold set at 0.05, and corrected in case of multiple group comparison. The study analysis was conducted by SPSS version 25 (IBM, Armonk, NY, USA).

## 4. Discussion

The main finding of the present study is that in patients ≥ 80 years with a high degree of frailty (based on a CFS value > 6), the PCT values at ED admission do not predict infective diagnosis, nor are associated with higher odds of in-hospital death. Still, in frail older adults, the PCT values in ED could be a useful predictor of bloodstream infection.

In patients with suspected infective disease in the ED, a timely diagnosis is critical for effective management and the timely administration of antibiotics to avoid the development of sepsis and septic shock [7]. A certain diagnosis is made by performing specific microbiological tests; the results of which, however, are not readily available in the ED [5].

The assessment of frailty with the CFS and the comprehensive multidimensional assessment of older patients combine to better characterize the suspicion of infection in this group. Older adults often enter the ED for symptoms that appear unrelated to the infectious event (e.g., falling, altered consciousness, dysphagia, weakness, dizziness, etc.), and the infectious diagnosis may be delayed [21,22]. On the other hand, it is also possible that localized or mild infections may trigger more significant clinical pictures in older patients, and may manifest themselves with atypical symptoms, e.g., reduced level of vigilance, psychomotor agitation, respiratory failure, and renal failure [23,24].

In the present study, fever was the most frequent symptom for patients with a confirmed infective diagnosis. However, its prevalence decreased as an initial presentation in the transition from fit to very frail patients, which also occurred for other typical (vomiting, abdominal pain) and atypical symptoms (fatigue). This finding is crucial, as it highlights that in older adults classified as very frail, diagnostic suspicion is more complex, as it can only be based to a small extent on apparent symptoms and what the patient can report [25,26]. These data are in agreement with what is known in the literature, where atypical presentations are significantly more frequent in elderly, institutionalized, and cognitively impaired patients [16,25]. In these patients, the available laboratory diagnostic tools may not be sufficient for an early diagnosis, and fever may be absent as an onset symptom in 20–30% of patients [23]. The infective diagnosis may be even more difficult in the case of physicians not familiar with the various non-classical manifestations of infections in this age group [25,26].

Apart from a better characterization of older patients, the CFS score has a fair accuracy in stratifying the risk of poor outcomes [27,28], and the utilization of these scores in the ED may be useful to identify elderly patients at risk and provide timely interventions. Concerning the CFS score, there is a great deal of scientific evidence that correlates frailty with major outcomes and, in particular, with intra-hospital mortality; this correlation concerns both internal and surgical diseases [29,30]. Specifically in infectious diseases, the recent pandemic period of SARS-CoV-2 infection has dramatically demonstrated the role of frailty in correlating prognosis and treatment choices [31].

In the ED, most patients with diagnostic suspicion of infection, regardless of age and frailty, are treated with broad-spectrum antibiotic therapy, with rising concerns for the development of bacterial resistance. In this context, the use of markers such as PCT address the need for reducing unnecessary and prolonged wide-spectrum antibiotic therapy. Nevertheless, the use of PCT in geriatric patients is still debated, as the diagnostic and prognostic benefits of its dosage for infection management are unclear [32,33,34,35]. This is particularly true for patients whose general health status and functional independence are more compromised (i.e., those with frailty or disability), where many factors may influence the blood levels of this marker. In elderly subjects, inflammation, immune senescence, and chronic inflammatory diseases are correlated with PCT synthesis even in the absence of infection, limiting the diagnostic and prognostic role of this biomarker [13]. For these reasons, the correlation between PCT, sepsis diagnosis, and prognosis is still uncertain for older patients [13,36,37].

The present analysis revealed that PCT values at the ED presentation were not independently associated with poor prognosis and infective diagnosis in severely frail patients, with the possible exception of bloodstream infections (Table 4). This finding is in line with previously published data [9]. The addition of the frailty variable provides new insights into the role of PCT evaluation for patients ≥ 80 years, and the predictive role of PCT cannot be considered indiscriminately valid for the entire elderly population. It could be speculated that the accuracy of PCT for the identification of infection only in fit/low-frailty elderly patients could find its explanation in the lower comorbidity of this group compared to very frail patients.

On the other hand, the reduced predictive power of qSOFA in very frail elderly patients is probably justified by the confounding effect given by the overlapping of different conditions of severity (advanced dementia, hypomobility, bed rest, sensory deprivation, and multisystem vasculopathy).

### Study Limitations

Although conducted in a large sample of patients ≥ 80 years, this study has some limitations. As this is a retrospective observational case–control study, it is not possible to draw definitive conclusions; longitudinal studies and possibly randomized controlled clinical trials are needed to confirm our findings. Additionally, this is a single-center observational study, so the results cannot be generalizable to all Emergency Departments, particularly to those that do not hold a dedicated geriatric unit.

## 5. Conclusions

Early frailty assessment in the ED for patients ≥ 80 years with suspicion of infective diagnosis could provide relevant clues to the clinical and laboratory evaluation of these patients.

The present data suggest that in very frail patients, the role of PCT for early stratification of infective diagnosis could be limited, with the possible exception of bloodstream infections. In these patients, the prognostic value of PCT for the prediction of in-hospital death is also reduced.

In consideration of the need to improve the use of available health and social resources, according to our data, the fit elderly population would seem to be the one that would benefit most from early PCT dosing in terms of identifying infection, maintaining the same predictive factor as in younger patients. In the very frail population, the poor prognosis predictive value of common clinical findings would suggest the worth of comprehensive second-level geriatric assessment to identify and accelerate any possible focused healthcare measures.

## Figures and Tables

**Figure 1 antibiotics-12-01036-f001:**
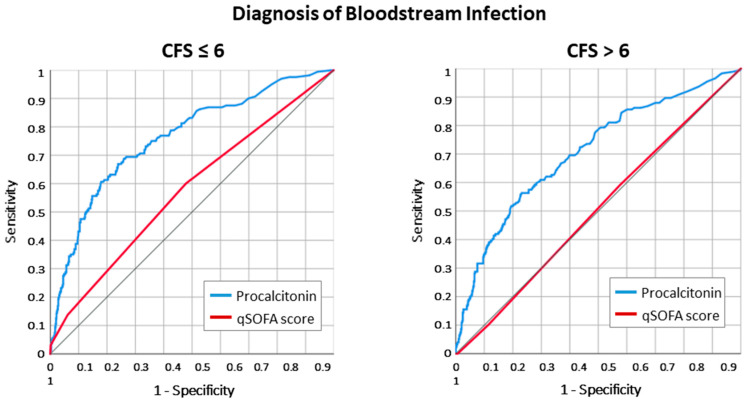
ROC curve analysis for the association with bloodstream infection diagnosis of procalcitonin (PCT) value and qSOFA score at the emergency department admission of adult patients ≥ 80 years old. In frail patients, the predicting power of PCT for bloodstream infections is slightly reduced compared to non- or moderately frail patients (ROC AUC 0.706 [0.660–0.753] vs. ROC AUC 0.762 [0.719–0.805], Z statistic = 1.720, *p* = 0.085 for the comparison).

**Figure 2 antibiotics-12-01036-f002:**
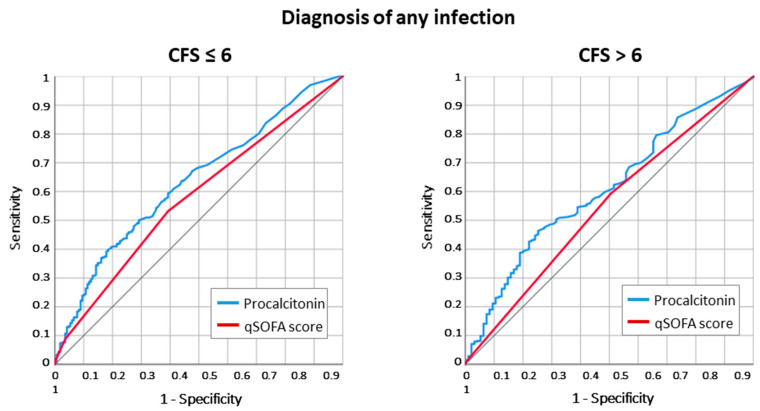
ROC curve analysis for the association with any infective diagnosis at hospital discharge in patients ≥ 80 years old. In frail patients, the predictive power of PCT at ED admission for infective diagnosis was reduced compared to non-/moderately frail patients (ROC AUC 0.607 [0.569–0.645] vs. ROC AUC 0.626 [0.601–0.651], Z statistic = 0.688, *p* = 0.491 for the comparison).

**Figure 3 antibiotics-12-01036-f003:**
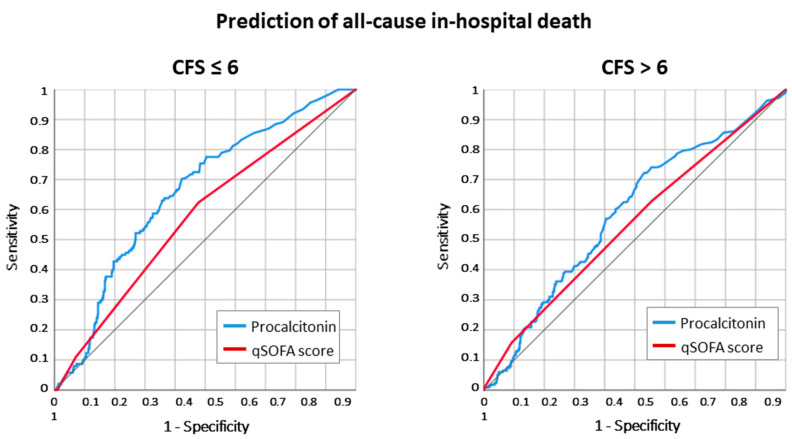
ROC curve analysis for the association with all-cause in-hospital death in patients ≥ 80 years old. In frail patients, the accuracy of PCT at ED admission was significantly reduced compared to non-/moderately frail patients (ROC AUC 0.586 [0.548–0.624] vs. ROC AUC 0.659 [0.625–0.692], Z statistic = 2.196, *p* = 0.028 for the comparison).

**Table 1 antibiotics-12-01036-t001:** The characteristics of patients divided according to frailty patterns.

	All Patients *n* 1459	Fit or Moderately Frail (CFS ≤ 6) *n* 796	Frail (CFS > 6) *n* 663	*p* Value
Age	85 (82–89)	85 (82–89)	86 (82–90)	0.014
Sex (Male)	718 (49.2)	428 (53.8)	290 (43.7)	<0.001
**ED Presentation**				
Triage				
-Emergency	269 (18.4)	121 (15.2)	148 (22.3)	
-Urgency	842 (57.7)	470 (59.0)	372 (56.1)	0.001
-Minor urgency	348 (23.9)	205 (25.8)	143 (21.6)	
qSOFA ≥ 2	136 (9.3%)	61 (7.7%)	75 (11.3%)	0.017
Dyspnea	540 (37.0)	285 (35.8)	255 (38.5)	0.295
Fever	706 (48.4)	413 (51.9)	293 (44.2)	0.003
Chest pain	62 (4.2)	40 (5.0)	22 (3.3)	0.108
Vomit	136 (9.3)	90 (11.3)	46 (6.9)	0.004
Abdominal pain	151 (10.3)	107 (13.4)	44 (6.6)	<0.001
Confusion	138 (9.5)	81 (10.2)	57 (8.6)	0.305
Malaise/fatigue	199 (13.6)	136 (17.1)	63 (9.5)	<0.001
**Laboratory values**				
Procalcitonin (ng/mL)	0.34 [0.13–1.61]	0.29 [0.12–1.53]	0.40 [0.15–1.92]	0.005
*Procalcitonin > 0.5 ng/mL*	624 (42.8%)	330 (41.5%)	294 (44.3%)	0.267
Hemoglobin (g/dl)	11.5 [10.1–12.7]	11.6 [10.2–12.8]	11.4 [9.9–12.6]	0.349
WBC (×10^9^/L)	11.3 [8.5–17.3]	11.2 [8.4–16.9]	11.3 [8.7–17.4]	0.725
Platelets (×10^9^/L)	223 [169–299]	211 [157–284]	239 [184–346]	0.017
Fibrinogen (mg/dL)	558 [432–765]	589 [404–808]	546 [437–729]	0.876
Creatinine (mg/dL)	1.32 [0.86–1.86]	1.32 [0.91–1.71]	1.28 [0.84–1.91]	0.634
Glucose (mg/dL)	129 [108–168]	122 [106–168]	133 [112–178]	0.054
CRP (mg/L)	121 [39–214]	112 [37–232]	123 [39–182]	0.527
**Comorbidities**				
Charlson Comorbidity Index	7 (6–9)	7 (6–9)	7 (6–9)	0.005
Hypertension	598 (41.0)	350 (44.0)	248 (37.4)	0.011
Ischemic heart disease	330 (22.6)	184 (23.1)	146 (22.0)	0.619
Congestive heart failure	570 (39.1)	310 (38.9)	260 (39.2)	0.916
Peripheral vascular disease	531 (36.4)	265 (33.3)	266 (40.1)	0.007
Cerebrovascular disease	197 (13.5)	84 (10.6)	113 (17.0)	<0.001
Dementia	249 (17.1)	73 (9.2)	176 (26.5)	<0.001
COPD	364 (24.9)	209 (26.3)	155 (23.4)	0.206
Diabetes	362 (24.8)	181 (22.7)	181 (27.3)	0.045
Liver chronic disease	34 (2.3)	20 (2.5)	14 (2.1)	0.613
Rheumatologic disease	28 (1.9)	19 (2.4)	9 (1.4)	0.154
Chronic kidney disease	538 (36.9)	291 (36.6)	247 (37.3)	0.783
Malignancy	295 (20.2)	170 (21.4)	125 (18.9)	0.236
**Site of infection**				
Any infection	1196 (82.0%)	628 (78.9%)	568 (85.7%)	0.001
Sepsis	334 (22.9)	160 (20.1)	174 (26.2)	0.005
Pneumonia	638 (43.7)	328 (41.2)	310 (46.8)	0.033
UTI	297 (20.4)	153 (19.2)	144 (21.7)	0.238
Abdominal infection	186 (12.7)	117 (14.7)	69 (10.4)	0.014
Others	79 (5.4)	46 (5.8)	33 (5.0)	0.501
**Outcomes**				
Length of stay	11.3 [7.2–17.7]	10.7 [7.3–17.2]	11.5 [7.2–18.5]	0.321
In-hospital death	354 (24.3%)	138 (17.3%)	216 (32.6%)	<0.001

CFS—Clinical Frailty Scale, WBC—white blood cells, CRP—C-reactive protein, COPD—chronic obstructive pulmonary disease, UTI—urinary tract infection.

**Table 2 antibiotics-12-01036-t002:** Factors associated with infection in the study cohort according to frailty status.

	Fit or Moderately Frail (CFS ≤ 6)			Frail (CFS > 6)		
	Infection (Any) *n* 628	Non-Infected *n* 168	*p*	Infection (Any) *n* 568	Non-Infected *n* 95	*p*
Age	85 [83–89]	85 [82–89]	0.249	86 [83–90]	85 [82–89]	0.159
Sex (Male)	341 (54.3)	87 (51.8)	0.562	253 (44.5)	37 (38.9)	0.309
CFS	5 [4–6]	5 [4–6]	0.526	8 [7,8]	7 [7–8]	0.011
**ED Presentation**						
Triage						
-Emergency	98 (15.6%)	23 (13.7)		130 (22.9%)	18 (18.9)	
-Urgency	368 (58.6)	102 (60.7)	0.810	315 (55.5)	57 (60.0)	0.644
-Minor urgency	162 (25.8)	43 (25.6)		123 (21.7)	20 (21.1)	
QSOFA ≥ 2	55 (8.8%)	6 (3.6)	0.025	66 (11.6)	9 (9.5)	0.541
Dyspnea	234 (37.3)	51 (30.4)	0.097	235 (41.4)	20 (21.1)	<0.001
Fever > 38 °C in ED	351 (55.9)	62 (36.9)	<0.001	268 (47.2)	25 (26.3)	<0.001
Chest pain	28 (4.5)	12 (7.1)	0.157	19 (3.3)	3 (3.2)	0.925
Vomiting	66 (10.5)	24 (14.3)	0.170	38 (6.7)	8 (8.4)	0.539
Abdominal pain	89 (14.2)	18 (10.7)	0.243	36 (6.3)	8 (8.4)	0.450
Diarrhea	48 (7.6)	9 (5.4)	0.307	39 (6.9)	8 (8.4)	0.585
Neurological sympt.	62 (9.9)	19 (11.3)	0.584	51 (9.0)	6 (6.3)	0.391
Malaise/fatigue	100 (15.9)	36 (21.4)	0.092	54 (9.5)	9 (9.5)	0.992
** *Laboratory values* **						
PCT (ng/mL)	0.39 [0.14–2.48]	0.18 [0.10–0.56]	<0.001	0.46 [0.16–2.45]	0.28 [0.10–0.58]	0.001
PCT > 0.5 ng/mL	287 (45.6)	43 (25.6)	<0.001	268 (47.2)	26 (27.4)	<0.001
Hemoglobin (g/dL)	11.8 [10.2–13.1]	11.1 [10.2–12.2]	0.196	11.2 [9.8–12.6]	11.8 [10.4–12.6]	0.559
WBC (×10^9^/L)	11.3 [8.5–17.3]	11.2 [9.0–13.9]	0.888	12.3 [8.8–18.5]	11.3 [8.6–17.4]	0.812
Platelets (×10^9^/L)	223 [169–299]	225 [155–292]	0.417	223 [174–351]	250 [225–343]	0.202
Fibrinogen (mg/dL)	591 [423–809]	588 [374–528]	0.296	548 [437–750]	533 [444–682]	0.852
Creatinine (mg/dL)	1.30 [0.88–1.68]	1.49 [1.21–4.37]	0.010	1.25 [0.83–1.90]	1.50 [0.90–2.29]	0.434
Glucose (mg/dL)	121 [105–166]	128 [106–188]	0.482	132 [115–180]	135 [105–188]	0.849
CRP (mg/L)	146 [56–234]	38 [18–155]	0.019	131 [41–86]	43 [12–177]	0.154
** *Comorbidities* **						
Charlson Index	7 [6–9]	7 [6–9]	0.258	7 [6–9]	8 [6–9]	0.101
Hypertension	276 (43.9)	74 (44.0)	0.982	210 (37.0)	38 (40.0)	0.572
IHD	146 (23.2)	38 (22.6)	0.864	126 (22.2)	20 (21.1)	0.806
CHF	244 (38.9)	66 (39.3)	0.919	218 (38.4)	42 (44.2)	0.281
PVD	208 (33.1)	57 (33.9)	0.844	222 (39.1)	44 (46.3)	0.183
Previous stroke	63 (10.0)	21 (12.5)	0.355	93 (16.4)	20 (21.1)	0.262
Dementia	59 (9.4)	14 (8.3)	0.672	144 (25.4)	32 (33.7)	0.089
COPD	180 (28.7)	27 (17.3)	0.003	144 (25.4)	11 (11.6)	0.003
Diabetes	145 (23.1)	36 (21.4)	0.648	152 (26.8)	29 (30.5)	0.446
Chronic liver disease	14 (2.2)	6 (3.6)	0.324	11 (1.9)	3 (3.2)	0.443
Rheumatologic	11 (1.8)	8 (4.8)	0.023	9 (1.6)	0 (0.0)	0.217
CKD	234 (37.3)	57 (33.9)	0.426	213 (37.5)	34 (35.8)	0.450
Malignancy	127 (20.2)	43 (25.6)	0.131	100 (17.6)	25 (26.3)	0.045
** *Outcomes* **						
Length of stay	11.3 [7.4–17.4]	9.5 [6.3–15.6]	0.012	11.6 [7.2–18.8]	11.4 [5.6–17.1]	0.298
Death	108 (17.2%)	30 (17.9%)	0.841	187 (32.9%)	29 (30.5%)	0.645

ED—emergency department, CFS—Clinical Frailty Scale, WBC—white blood cells, CRP—C-reactive protein, COPD—chronic obstructive pulmonary disease, IHD—ischemic heart disease, CHF—congestive heart failure, PVD—peripheral vascular disease, CKD—chronic kidney disease.

**Table 3 antibiotics-12-01036-t003:** Factors associated with death in the study cohort according to frailty status.

	Fit or Moderately Frail (CFS ≤ 6)			Frail (CFS ≥ 6)		
	Deceased *n* 138	Alive *n* 658	*p* Value	Deceased *n* 216	Alive *n* 447	*p* Value
**Variable**						
Age	86 [83–90]	85 [82–89]	0.108	86 [83–91]	86 [82–90]	0.250
Sex (Male)	71 (51.4)	357 (54.3)	0.548	102 (47.2)	188 (42.1)	0.209
Clinical Frailty Scale	5 (5–6)	5 (4–6)	0.054	8 (7–8)	7 (7–8)	0.055
**ED Presentation**						
Triage						
-Emergency	33 (23.9%)	88 (13.4)		74 (34.3%)	74 (16.6)	
-Urgency	77 (55.8)	393 (59.7)	0.005	108 (50.0)	264 (59.1)	<0.001
-Minor urgency	28 (20.3)	177 (26.9)		34 (15.7)	109 (24.4)	
QSOFA ≥ 2	15 (10.9%)	46 (7.0%)	0.119	34 (15.7%)	41 (9.2%)	0.012
Dyspnea	62 (44.9)	223 (33.9)	0.014	86 (39.8)	169 (37.8)	0.619
Fever	57 (41.3)	356 (54.1)	0.006	81 (37.5)	212 (47.4)	0.016
Chest pain	6 (4.3)	34 (5.2)	0.689	4 (1.9)	18 (4.0)	0.143
Vomiting	16 (11.6)	74 (11.2)	0.907	14 (6.5)	32 (7.2)	0.748
Abdominal pain	12 (8.7)	95 (14.4)	0.072	9 (4.2)	35 (7.8)	0.076
Diarrhea	15 (10.9)	42 (6.4)	0.063	10 (4.6)	37 (8.3)	0.086
Neurological symptoms	14 (10.1)	67 (10.2)	0.989	13 (6.0)	44 (9.8)	0.100
Malaise/fatigue	24 (17.4)	112 (17.0)	0.916	21 (9.7)	42 (9.4)	0.893
**Laboratory values**						
Procalcitonin (ng/mL)	1.11 [0.27–4.29]	0.26 [0.12–1.38]	<0.001	0.57 [0.22–3.34]	0.31 [0.14–0.26]	<0.001
Procalcitonin > 0.5 ng/mL	88 (63.8%)	242 (36.8%)	<0.001	117 (54.2%)	177 (39.6%)	<0.001
Hemoglobin (g/dL)	11.5 [10.2– 12.7]	12.3 [10.2–13.4]	0.170	10.6 [8.3–11.6]	11.8 [10.2–12.7]	0.002
WBC (×10^9^/L)	14.2 [7.7–18.3]	11.0 [8.4–16.7]	0.466	11.3 [8.4–17.4]	12.3 [9.2–17.5]	0.459
Platelets (×10^9^/L)	225 [179–291]	208 [155–284]	0.279	220 [169–368]	248 [189–343]	0.574
Fibrinogen (mg/dL)	600 [398–770]	588 [404–810]	0.900	551 [453–731]	533 [415–722]	0.482
Creatinine (mg/dL)	1.38 [0.8–3.24]	1.32 [0.92–1.69]	0.544	1.45 [1.0–1.94]	1.16 [0.78–1.89]	0.208
Glucose (mg/dL)	120 [106–148]	122 [106–170]	0.676	135 [102–173]	133 [115–182]	0.628
CRP (mg/L)	226 [53–258]	107 [33–218]	0.107	131 [36–188]	121 [39–184]	0.736
**Clinical History**						
Charlson Comorbidity Index	7 (6–9)	7 (6–8)	0.551	7 (6–9)	8 (7–8)	0.023
Hypertension	51 (37.0)	299 (45.4)	0.068	68 (31.5)	180 (40.3)	0.028
IHD	38 (27.5)	146 (22.2)	0.175	54 (25.0)	92 (20.6)	0.198
CHF	63 (45.7)	247 (37.5)	0.076	103 (47.7)	157 (35.1)	0.002
PVD	52 (37.7)	213 (32.4)	0.229	86 (39.8)	180 (40.3)	0.911
Previous stroke	18 (13.0)	66 (10.0)	0.295	31 (14.4)	82 (18.3)	0.200
Dementia	16 (11.6)	57 (8.7)	0.278	42 (19.4)	134 (30.0)	0.004
COPD	30 (21.7)	179 (27.2)	0.185	48 (22.2)	107 (23.9)	0.625
Diabetes	32 (23.2)	149 (22.6)	0.890	61 (28.2)	120 (26.8)	0.706
Liver chronic disease	4 (2.9)	16 (2.4)	0.750	1 (0.5)	13 (2.9)	0.040
Rheumatologic	3 (2.2)	16 (2.4)	0.857	0 (0.0)	9 (2.0)	0.036
Chronic kidney disease	56 (40.6)	235 (35.7)	0.281	76 (35.2)	171 (38.3)	0.444
Malignancy	36 (26.1)	134 (20.4)	0.136	39 (18.1)	86 (19.2)	0.715
**Outcomes—Site of Infection**						
Sepsis	37 (26.8)	123 (18.7)	0.030	90 (41.7)	84 (18.8)	<0.001
Pneumonia	72 (52.2)	256 (38.9)	0.004	101 (46.8)	209 (46.8)	0.999
UTI	9 (6.5)	144 (21.9)	<0.001	26 (12.0)	118 (26.4)	<0.001
Abdominal infection	8 (5.8)	109 (16.6)	0.001	14 (6.5)	55 (12.3)	0.021
Others	5 (3.6)	41 (6.2)	0.233	14 (6.5)	19 (4.3)	0.216
Length of stay (days)	10.3 [4.36–18.00]	11.01 [7.42–7.19]	0.043	8.74 [3.65–6.29]	12.6 [8.30–19.59]	<0.001

ED—emergency department, CFS—Clinical Frailty Scale, WBC—white blood cells, CRP—C-reactive protein, COPD—chronic obstructive pulmonary disease, IHD—ischemic heart disease, CHF—congestive heart failure, PVD—peripheral vascular disease, UTI—urinary tract infection.

**Table 4 antibiotics-12-01036-t004:** Multivariate logistic regression models for the association of procalcitonin value with infective diagnosis or death in frail patients compared to controls. Constant was included in all the logistic models.

**Prediction of Bloodstream Infection—Non-Frail or Moderately Frail Patients (CFS ≤ 6)**			
**Variable**	**Wald Statistic**	**Odds Ratio [95% CI]**	***p* Value**
PCT values	44.413	1.06 [1.04–1.08]	**<0.001**
Age	1.616	0.97 [0.93–1.01]	0.204
qSOFA ≥ 2	10.038	2.52 [1.42–4.46]	**0.002**
Charlson index	0.0966	1.01 [0.93–1.09]	0.756
**Prediction of bloodstream infection—Very Frail patients (CFS > 6)**			
PCT values	31.694	1.05 [1.03–1.07]	**<0.001**
Age	4.482	0.96 [0.92–0.99]	**0.034**
qSOFA ≥ 2	1.311	0.70 [0.38–1.29]	0.252
Charlson index	6.130	0.91 [0.84–0.98]	**0.013**
**Prediction of any infection—Non-frail or moderately frail patients (CFS ≤ 6)**			
PCT values	6.798	1.04 [1.01–1.08]	**0.009**
Age	1.719	1.03 [0.99–1.07]	0.190
qSOFA ≥ 2	4.653	2.59 [1.09–6.14]	**0.031**
Charlson index	0.327	0.98 [0.91–1.05]	0.568
**Prediction of any infection—Very Frail patients (CFS > 6)**			
PCT values	2.687	1.02 [0.99–1.06]	0.130
Age	2.293	1.04 [0.99–1.09]	0.734
qSOFA ≥ 2	0.658	1.14 [0.54–2.38]	0.734
Charlson index	0.327	0.97 [0.88–1.05]	0.417
**Prediction of all-cause death—Non-frail or moderately frail patients (CFS ≤ 6)**			
Procalcitonin value	2.599	1.01 [1.00–1.02]	**0.047**
Age	0.895	1.03 [0.99–1.07]	0.118
Urgency	10.332	2.28 [1.28–4.06]	**0.006**
Emergency	7.766	1.09 [0.68–1.77]	**0.005**
Non-Urgency	0.148	Reference	0.701
Charlson index	0.767	1.13 [1.04–1.22]	**0.002**
**Prediction of all-cause death—Very Frail patients (CFS > 6)**			
Procalcitonin value	0.004	1.00 [0.98–1.02]	0.948
Age	2.446	1.03 [0.99–1.07]	0.118
Emergency	23.576	2.99 [1.79–4.99]	**<0.001**
Urgency	17.640	1.26 [0.80–1.98]	**<0.001**
Non-Urgency	0.999	Reference	0.318
Charlson index	1.003	0.97 [0.90–1.03]	0.317

## Data Availability

The data presented in this study are available upon reasonable request to the corresponding author.

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
