# Peer review of "Reduced Prognostic Role of Serum PCT Measurement in Very Frail Older Adults Admitted to the Emergency Department"

_antibiotics, 2023, doi:10.3390/antibiotics12061036_

Round 1
Reviewer 1 Report
A lot of articles are devoted to the relationship between inflammatory parameters and frailty in old people. For example:
RT Dogrul et al. (doi:10.4274/ejgg.galenos.2019.29) - FS / ESR correlation,
P Gomez-Rubio et al. (doi:10.3390/diagnostics 12010117) – FS / IL-6 correlation,
P Soysal et al: (doi.erg/10.1016/j.eurr.2016.03.006) meta-analysis FS / CRP,IL-6,
and other. This work complements the FS / PCT correlation in a suitable way.
Author Response
Dear reviewer,
thank you so much for your comment.
We have made some changes, at your suggestion, and we hope it will make the work better.
About your note, we completely agree. We have proceeded to improve this aspect.
Best wishes.
Reviewer 2 Report
I have read your study. It is well designed and very interesting. Figures and tables are clear. Regarding tables , I suggest to correct distances among columns.
Author Response

(The authors gave the same response as above.)

Reviewer 3 Report
This article is very interesting, diagnostic accuracy being calculated by ROC analysis and compared in the patients with severe frailty, measured by CFS >6 . The PCT values in ED could be a useful predictor of bloodstream infection. These values at ED admission do not predict infective diagnosis, nor associated with higher odds of in-hospital death.
Author Response
Dear reviewer,
thank you so much for your comment.
We have made some changes, at your suggestion, and we hope it will make the work better.
Best wishes.
Reviewer 4 Report
Dear authors.
I find it that this manuscript is interesting and may attract attention from many readers.
However, there are some points to be addressed.
Abstract
1. In the method, I don’t think this study is a cross-sectional study.; this study were conducted using data for 6 year periods.
2. In the results, you need to provide ORs and its 95 CI% for the results you described.
Introduction
1. I think the Introduction section is long, so it needs to be shortened.
- In the first paragraph, the sentences in line 50-54 can be deleted.
- The 4th paragraph seems also unnecessary. You can delete it.
2. The qSOFA is quick SOFA, not rapid SOFA.
Methods
1. As abovementioned, I don’t think it is cross-sectional study.
2. The inclusion criteria include elderly patients aged > 80 years and clinical suspicion of infection, but non-infected patients were also included in your study results. Hence, I think you need to revise the inclusion criteria.
3. What is the evidence for dividing patients into two groups, fit/moderately frail (CFS ≤6) vs. frail groups (CFS >6).
4. In the 2.2. Study variables, you described that the cut-off value of PCT serum level predictive of sepsis was set at 0.5 ng/ml. I thinks the threshold of 0.5 ng/ml is for ‘infection’ not ‘sepsis’.
5. In the Outcome Variables, I think the primary endpoint is ‘predictive ability of PCT for in-hospital mortality’ rather than ‘all-cause hospital mortality’.
6. About the secondary outcomes, you need to elaborate on the definitions for bloodstream infection (BSI) and any infections (e.g., pneumonia, UTI, abdominal infections). For the diagnosis of BSI, at least two blood samples with positive results are needed, and for other infections, an isolation of organisms in biological samples is not sufficient for the diagnosis of infection. This is important to make your study more like a scientific article.
7. Regarding infections confirmed at the hospital discharge, do you mean this ‘any infections during the hospitalization’? You need to clarify this because new infections occurred during the later period of hospitalizations are not associated with initial PCT levels measured at EDs, which could bias your results.
Results
1. All the tables (Table 1-4) looks too large. It may prevent readers to understand.
2. There are no footnotes in all tables. You need to add footnotes where full spellings (letters) for abbreviations are described.
3. There are no units for figures (numbers) in the main text and all tables. Please add units in all the figures (numbers).
3. In the ‘2.1. Infection diagnosis according to the frailty status’ and Table 1, I think you use ‘sepsis’ and ‘bloodstream infection’ concomitantly (interchangeably). You need to clarify this. As you know, bloodstream infection is different from sepsis.
4. Table 2 and 3, the threshold of ‘PCT < 0.05 ng/ml’ seems wrong. Is it ‘PCT < 0.5 ng/ml’?
5. In the Figure 1 and 2 legends, the ‘accuracy’ should be replaced with other words (e.g., predicting power or discrimination power) because the accuracy has its own meaning and defining formula in the ROC curves.
Discussion
1. The Discussion section is too long. So, I hope that you can shorten it.
- The 4th and 5th paragraphs (line 332 – 349) can be deleted from the Discussion.
- It looks like that most sentences in the 7th paragraph is not about your study. Please revise them if possible.
2. Most importantly, the Discussion should be focused on your study. For example, any methodological problems, some differences between your data and those of previous studies and its reasons, and clinical implications of your study results.
Thank you.
None
Author Response
Dear reviewer,
thank you very much for your advice that will help us to improve our work.
Trying to answer point by point:
About Abstract:
1)In the method, I don’t think this study is a cross-sectional study; this study were conducted using data for 6 year periods.
Thank you for the note. It was a typo. This is an observational study. We correct the abstract and the methods section.
2)In the results, you need to provide ORs and its 95 CI% for the results you described.
Grateful for the observation. We completely agree. We have proceeded to improve this aspect.
About Introduction:
1)I think the Introduction section is long, so it needs to be shortened.
- In the first paragraph, the sentences in line 50-54 can be deleted
- The 4th paragraph seems also unnecessary. You can delete it.
Thank you for the remark. We agree with your suggestion. We have removed the excess parts (sentences in line 50-54 and the 4th paragraph) and adjusted the bibliographical references.
2)The qSOFA is quick SOFA, not rapid SOFA
Thank you for the note, It was a typo, we promptly corrected it.
About Methods:
1)As abovementioned, I don’t think it is cross-sectional study.
Thank you for the note. This is an observational study. We correct the abstract and the methods section. It was a typo.
2)The inclusion criteria include elderly patients aged > 80 years and clinical suspicion of infection, but non-infected patients were also included in your study results. Hence, I think you need to revise the inclusion criteria.
Thank you for the question. Clinical suspicion of infection was based on the value of PCT biomarker and qSOFA score, later diagnostic imaging and culture tests could confirm or not confirm the presence of infection. The presence of the group of 'uninfected' patients is subsequently useful in the discussion of the article.
3)What is the evidence for dividing patients into two groups, fit/moderately frail (CFS ≤6) vs. frail groups (CFS >6).
Thank you for the question. We decided to stratify patients in this way to put more emphasis on older age (age > or equal to 80 years) and severe frailty (CFS > 6) using the Comprehensive Geriatric Assessment at the ‘front door’ of our Emergency Department.
4)In the 2.2. Study variables, you described that the cut-off value of PCT serum level predictive of sepsis was set at 0.5 ng/ml. I thinks the threshold of 0.5 ng/ml is for ‘infection’ not ‘sepsis’
Thank you for the remark. We totally agree. We have proceeded to correct it.
5)In the Outcome Variables, I think the primary endpoint is ‘predictive ability of PCT for in-hospital mortality’ rather than ‘all-cause hospital mortality’.
Grateful for the suggestion. We completely agree. We have proceeded to fix this aspect too.
6)About the secondary outcomes, you need to elaborate on the definitions for bloodstream infection (BSI) and any infections (e.g., pneumonia, UTI, abdominal infections). For the diagnosis of BSI, at least two blood samples with positive results are needed, and for other infections, an isolation of organisms in biological samples is not sufficient for the diagnosis of infection. This is important to make your study more like a scientific article.
Thank you for the note. We agree with you. We have proceeded to improve this aspect.
- Regarding infections confirmed at the hospital discharge, do you mean this ‘any infections during the hospitalization’? You need to clarify this because new infections occurred during the later period of hospitalizations are not associated with initial PCT levels measured at EDs, which could bias your results.
Thank you for the question. The infections considered are those for which culture test results performed in the emergency department became positive during the stay in the ED. We do not refer to new infections that occurred during hospitalization.
About Results
1)All the tables (Table 1-4) looks too large. It may prevent readers to understand.
Thank you for the note. We have proceeded to solve this aspect.
- There are no footnotes in all tables. You need to add footnotes where full spellings (letters) for abbreviations are described.
Thank you for the note. We have proceeded to solve this aspect.
- There are no units for figures (numbers) in the main text and all tables. Please add units in all the figures (numbers).
Thank you for the note. We have proceeded to solve this aspect.
- In the ‘2.1. Infection diagnosis according to the frailty status’ and Table 1, I think you use ‘sepsis’ and ‘bloodstream infection’ concomitantly (interchangeably). You need to clarify this. As you know, bloodstream infection is different from sepsis.
Thank you for your report, it was a typo. We have corrected it.
- Table 2 and 3, the threshold of ‘PCT < 0.05 ng/ml’ seems wrong. Is it ‘PCT < 0.5 ng/ml’?
Thank you for your report, it was a typo. We have corrected it.
- In the Figure 1 and 2 legends, the ‘accuracy’ should be replaced with other words (e.g., predicting power or discrimination power) because the accuracy has its own meaning and defining formula in the ROC curves.
Thanks for the report, we agree with the suggestion. We have changed the legend.
About Discussion
- The Discussion section is too long. So, I hope that you can shorten it.
- The 4th and 5th paragraphs (line 332 – 349) can be deleted from the Discussion.
- It looks like that most sentences in the 7th paragraph is not about your study. Please revise them if possible.
Thanks for the suggestion, we have taken care to reduce the text.
The 7th paragraph refers to the presenting symptoms in our sample, which are in line with what is widely presented in the literature regarding the pattern of presentation of infectious diseases in elderly patients.
- Most importantly, the Discussion should be focused on your study. For example, any methodological problems, some differences between your data and those of previous studies and its reasons, and clinical implications of your study results.
Thank you very much for your suggestion and we agree with you. However, due to problems of length, and given the extensiveness of the topic, we have difficulty implementing the discussion further.
Best wishes
Reviewer 5 Report
1. Abstract of the manuscript is too long and extraordinarily descriptive. The authors should be concise in the abstract.
2. Its strongly suggested to provide the complete name before using any abbreviation e.g., ED. Refer to line number 18.
3. The title of the manuscript states the reduced prognostic value of procalcitonin (PCT) while in the abstract authors mention that “Conversely, the PCT values were confirmed to be a good independent predictor of bloodstream infection both in frail and controls” (line 35-36). Could authors like to explain this discrepancy?
4. PCT has a central role in this study. At the same time, in the introduction, the authors have not provided the basic mechanism or homeostasis of PCT release and its affiliation with sepsis and frail older adults.
5. Refer to line 89, “clinical suspicion of infection” authors need to provide detail in the manuscript.
6. It’s also suggested that authors must explicitly mention the type of infection rather than simply using the word infection e.g., bacterial infections vs viral infection etc.
7. Why have authors only included patients with ages≥ 80 years? What were the scientific reasons behind this?
8. Refer to line 98, “In the case of several measurements, the first values were considered” Clarify this statement.
9. Authors should adjust the line spacing of Tables 1, 2 & 3 and try to provide the data in a well-presentable manner. So, the reader can get crisp information rather than moving to the next pages for correlating the info.
10. Figures 1, 2, and 3 are not labeled.
11. In the end, a similar type of study is already reported and published by another Italian research group (https://www.mdpi.com/2079-6382/10/7/788), How do authors compare their findings with already published data?
There are several grammatical errors and some spelling mistakes.
Author Response
Dear reviewer,
thank you very much for your advice that will help us to improve our work.
Trying to answer point by point:
- Abstract of the manuscript is too long and extraordinarily descriptive. The authors should be concise in the abstract.
Thanks for the suggestion, we have taken care to reduce the text.
- Its strongly suggested to provide the complete name before using any abbreviation e.g., ED. Refer to line number 18.
Thanks for the suggestion, we had specified it in the text but not in the abstract. We have corrected it.
- The title of the manuscript states the reduced prognostic value of procalcitonin (PCT) while in the abstract authors mention that “Conversely, the PCT values were confirmed to be a good independent predictor of bloodstream infection both in frail and controls” (line 35-36). Could authors like to explain this discrepancy?
Thanks for the request, we agree with you. We have tried to make it clearer by editing the title, the abstract and the manuscript (paragraph 3.3).
- PCT has a central role in this study. At the same time, in the introduction, the authors have not provided the basic mechanism or homeostasis of PCT release and its affiliation with sepsis and frail older adults.
Thank you for the advice, we fully agree. We have implemented the introduction accordingly.
- Refer to line 89, “clinical suspicion of infection” authors need to provide detail in the manuscript.
Thanks for the suggestion, we agree with you. With 'clinical suspicion' we mean the presence of clinical signs and symptoms suggestive of infection. We have modified in the manuscript.
- It’s also suggested that authors must explicitly mention the type of infection rather than simply using the word infection e.g., bacterial infections vs viral infection etc.
Thank you for the note. The infections considered are bacterial infections. We specified this in line number 93.
- Why have authors only included patients with ages≥ 80 years? What were the scientific reasons behind this?
Thank you for the question. We decided to stratify patients in this way to put more emphasis on older age (age > or equal to 80 years) and severe frailty (CFS > 6) using the Comprehensive Geriatric Assessment at the ‘front door’ of our Emergency Department.
- Refer to line 98, “In the case of several measurements, the first values were considered” Clarify this statement.
Thank you for the question. During the length of stay in the emergency room, multiple PCT measurements are often taken in order to assess the trend of therapy. In this case, we considered the value at admission.
- Authors should adjust the line spacing of Tables 1, 2 & 3 and try to provide the data in a well-presentable manner. So, the reader can get crisp information rather than moving to the next pages for correlating the info.
Thank you for the note. We completely agree. We have proceeded to improve this aspect.
- Figures 1, 2, and 3 are not labeled.
Thank you for the note. We completely agree. We have proceeded to improve this aspect.
- In the end, a similar type of study is already reported and published by another Italian research group (https://www.mdpi.com/2079-6382/10/7/788), How do authors compare their findings with already published data?
Thank you for the note. The work in question is mentioned in the introduction, however we agree that it should also be included in the discussion. We referred to this in line 361.
Best wishes
Round 2
Reviewer 4 Report
Dear authors
Thank you for your efforts on the revision process.
There seems no further issues on the revised version.
Thank you
Author Response
Dear Reviewer,thank you very much for your feedback.
We are pleased that we were able to reply in a satisfactory manner.
Best regards
Reviewer 5 Report
The graphs presented in Figures 1,2 and 3 are of poor resolution and must be replaced with high resolution graphs.
There are only a few minor English errors.
Author Response
Dear Reviewer
thank you very much for your feedback. We are happy to have been able to reply comprehensively to your previous comments.
We have improved the resolution of the graphs and rechecked the text, correcting errors. We hope we have improved the work further,
Best regards